# Peptides derived from the SARS-CoV-2 receptor binding motif bind to ACE2 but do not block ACE2-mediated host cell entry or pro-inflammatory cytokine induction

Amit Mahindra[1], Gonzalo Tejeda[2], Mario Rossi[2,3], Omar Janha[2], Imogen Herbert[4], Caroline Morris[1], Danielle C. Morgan[1], Wendy Beattie[5], Augusto C. Montezano[5], Brian Hudson[2], Andrew B. Tobin[2], David Bhella[4], Rhian M. Touyz[5], Andrew G. Jamieson[1], George S. Baillie[5], Connor M. Blair[5]*

1 School of Chemistry, University of Glasgow, Glasgow, United Kingdom, 2 Institute of Molecular Cell & Systems Biology, School of Medical, Veterinary and Life Sciences, University of Glasgow, Glasgow, United Kingdom, 3 Department of Biotechnological and Applied Clinical Sciences, University of L'Aquila, L'Aquila, Italy, 4 MRC Centre for Virus Research, University of Glasgow, Glasgow, United Kingdom, 5 Institute of Cardiovascular and Medical Sciences, Medical, Veterinary and Life Sciences, University of Glasgow, Glasgow, United Kingdom

* connor.blair@glasgow.ac.uk

Data Availability Statement: All relevant data are within the manuscript and its Supporting Information files.

## Abstract

SARS-CoV-2 viral attachment and entry into host cells is mediated by a direct interaction between viral spike glycoproteins and membrane bound angiotensin-converting enzyme 2 (ACE2). The receptor binding motif (RBM), located within the S1 subunit of the spike protein, incorporates the majority of known ACE2 contact residues responsible for high affinity binding and associated virulence. Observation of existing crystal structures of the SARS-CoV-2 receptor binding domain ($S_{RBD}$)–ACE2 interface, combined with peptide array screening, allowed us to define a series of linear native RBM-derived peptides that were selected as potential antiviral decoy sequences with the aim of directly binding ACE2 and attenuating viral cell entry. RBM1 (16mer): S443KVGGNYNYLYRLFRK458, RBM2A (25mer): E484GFNCYFPLQSYGFQPTNGVGYQPY508, RBM2B (20mer): F456NCYFPLQSYGFQPTNGVGY505 and RBM2A-Sc (25mer): NYGLQGSPFGYQETPYPFCNFVQYG. Data from fluorescence polarisation experiments suggested direct binding between RBM peptides and ACE2, with binding affinities ranging from the high nM to low µM range ($K_d$ = 0.207–1.206 µM). However, the RBM peptides demonstrated only modest effects in preventing $S_{RBD}$ internalisation and showed no antiviral activity in a spike protein trimer neutralisation assay. The RBM peptides also failed to suppress S1-protein mediated inflammation in an endogenously expressing ACE2 human cell line. We conclude that linear native RBM-derived peptides are unable to outcompete viral spike protein for binding to ACE2 and therefore represent a suboptimal approach to inhibiting SARS-CoV-2 viral cell entry. These findings reinforce the notion that larger biologics (such as soluble ACE2, 'miniproteins', nanobodies and antibodies) are likely better suited as SARS-CoV-2 cell-entry inhibitors than short-sequence linear peptides.

**Funding:** Acknowledgements: CMB, MR, GT and OJ are supported through the WT ISSF COVID Response Award (204820/Z/16/Z). AM is supported through the EPSRC (Research Project Grant EP/N034260/2) and CM is supported through Dstl (DSTLX-1000141308). DCM thanks the EPSRC for a studentship (EP/N509668/1 and EP/R513222/1). RMT is supported through a BHF Chair award (CH/4/29762). ACM is supported by a University of Glasgow Walton Fellowship. DB and IH were supported by the Medical Research Council (MC UU 12014/7). Author Contributions: CMB and AM designed the study. CMB, AM, GT, MR, OJ, IH, DCM, CM, WB and ACM performed the experiments. CMB, AM and GSB wrote the manuscript. ACM, BH, ABT, DB, RMT, AGJ and GSB provided critical support and supervision of study. All authors read and approved manuscript.

**Competing interests:** The authors declare no competing interests.

# Introduction

The ongoing COVID19 (SARS-CoV-2) pandemic has resulted in > 235 million confirmed cases and > 4.8 million deaths globally since its outbreak in 2019 [1]. SARS-CoV-2 is a positive sense single stranded RNA virus classified within the betacoronavirus genus, primarily entering the host via epithelial cells within the upper respiratory tract [2]. Host cell invasion of SARS-CoV-2, similar to SARS-CoV-1, relies upon the direct interaction between its spike glycoprotein (S) and the host cell membrane bound angiotensin converting enzyme 2 (ACE2) [3, 4]. Key contact residues within the receptor binding domain (RBD) of SARS-CoV-2 (located within the S1 subunit of the S) play a critical role in mediating high affinity, low nM binding to ACE2. This association drives high viral infection rates and consequential pathogenic severity [5–8]. Neutralising antibodies raised against the RBD, soluble recombinant ACE2 (sACE2), and other related proteins have demonstrated a clear ability to block the RBD–ACE2 protein-protein interaction (PPI) and attenuate viral entry of both SARS-CoV-1 and SARS-CoV-2 strains [9]. These findings highlight the potential of disrupting the RBD–ACE2 protein-protein interaction (PPI) as a promising antiviral therapeutic approach.

Therapies capable of selectively disrupting PPIs represent a highly efficacious approach to treating certain diseases, with several demonstrating clear therapeutic benefit clinically [10]. However, effectively disrupting a PPI is highly challenging. The featureless interface of a PPI typically spans a large surface area (1,500–3,000 Å$^2$), consisting of multiple non-continuous binding sites (also referred to as 'hot spots') and no obvious deep binding pocket(s) [11]. This often renders traditional drug discovery approaches, such as competitive small molecules, a suboptimal approach to disrupting a given PPI due to their inability to exploit large surface areas (< 1,000 Å$^2$) [12]. Accordingly, development of PPI inhibitors has shifted focus to larger biologic molecules such as peptides (known as decoy, disruptor or interference peptides) [13]. Decoy peptides are typically designed from native structures containing a given PPI hot spot (s), meaning they possess high selectivity and affinity for their target(s), whilst constituting a low immunogenic risk [14, 15].

Research aimed at developing viral entry peptide inhibitors by competitively targeting the SARS-CoV-2 –ACE2 molecular interaction, has largely focussed on the antiviral potential of ACE2 sequence derived peptides/peptidomimetics [9]. These studies utilised existing co-crystal structures of ACE2 in complex with SARS-CoV-2 $S_{RBD}$, to design peptides that contain known RBD contact residues found within the α1 helix of the ACE2 binding interface [16–18]. Resulting antiviral efficacy of these ACE2 peptides varies considerably, depending largely on their size, stability, and secondary structure(s). Conversely, few studies have investigated peptide mimics derived from the SARS-CoV-2 $S_{RBD}$ sequence as potential antiviral PPI decoys. Notably, co-crystal structures of the SARS-CoV-2 RBD–ACE2 PPI highlight that all known contact residues within ACE2 are incorporated within the receptor binding motif (RBM, N437–Y508) of the S1 subunit [7]. Considering the dense location of these known ACE2-interacting hot spots clusters (Fig 1), the RBM represents a significant region for the rational development of short sequence RBD–ACE2 decoy peptides. Previous work utilising a SARS-CoV-1 spike RBM derived hexapeptide (Y438-K-Y-R-Y-L443) demonstrated a modest ability to directly bind ACE2 ($K_d$ = 46 μM) and inhibit coronavirus NL63 cell entry and subsequent viral replication [19]. Previous work utilising a SARS-CoV-1 spike RBM derived hexapeptide (Y438-K-Y-R-Y-L443) demonstrated a modest ability to directly bind ACE2 ($K_d$ = 46 μM) and inhibit coronavirus NL63 cell entry and subsequent viral replication [19]. Although this peptide was not tested against SARS-CoV-1 viral entry, these findings highlight the possibility for a SARS-CoV-2 RBM-derived peptide to represent a potentially promising approach to the development of novel antiviral decoy peptides. However, as the spike protein corresponding to

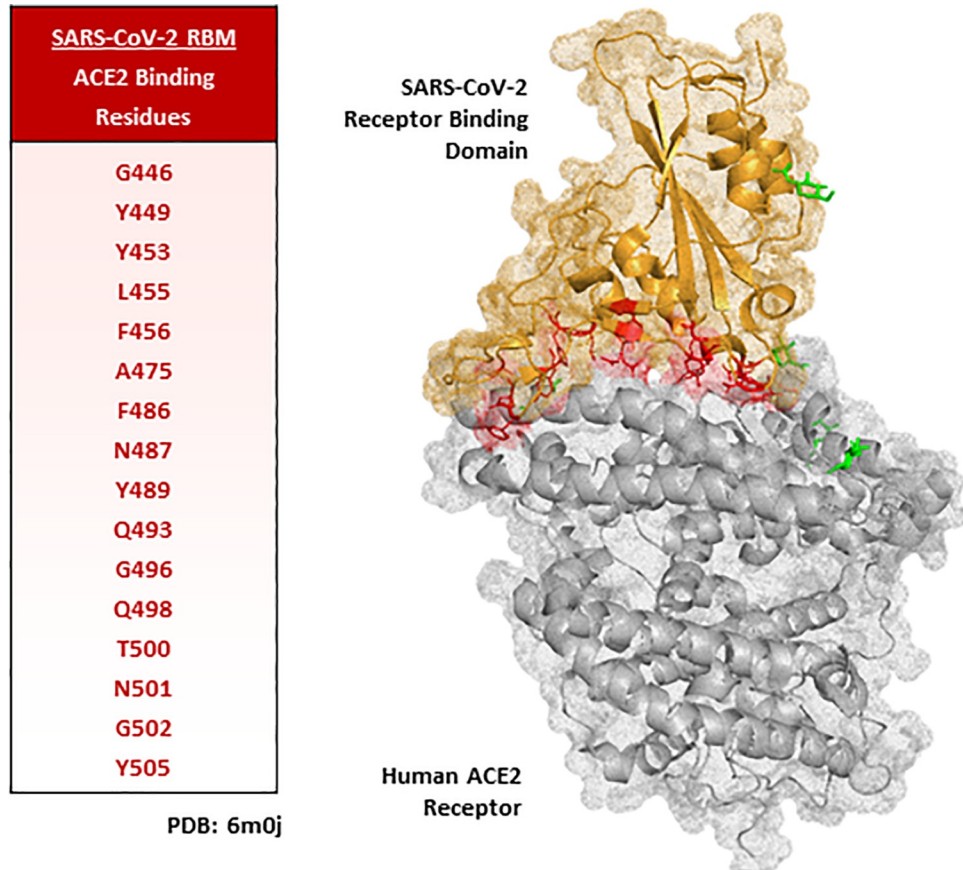

**Fig 1. SARS-CoV-2 receptor binding motif–ACE2 protein-protein interaction binding interface.** Amino acids in red highlight SARS-CoV-2 receptor binding motif (RBM) contact residues with ACE2.–adapted from (Lan, et al. 2020) [7]. ACE2 receptor–Grey. SARS-CoV-2 RBD–Orange. Green structures represent glycoproteins. Structure generated using PyMol software.

SARS-CoV-2 has been shown to bind ACE2 10 times more tightly than SARS-CoV-1, the binding affinity of SARS-CoV-2 derived peptide(s) will need to be significantly higher in order to be competitive against the spike protein [20].

In a unique approach, we have combined peptide array screening and existing co-crystal structures of the SARS-CoV-2 RBD–ACE2 PPI to select a novel set of peptides with potential anti-viral activity. We report on studies that evaluate the peptides' ability to (i) bind ACE2 directly, (ii) disrupt spike protein mediated internalisation, and (iii) inhibit spike protein induced pro-inflammatory cytokine production in human coronary microvascular epithelial cells (CMEC).

## Materials and methods

### Peptide array

Peptide experiments were performed as described previously [21]. SARS-CoV-2 receptor binding motif (RBM) peptides were produced by automatic SPOT synthesis [22, 23]. Peptides were synthesised on continuous cellulose membrane supports using 9-fluorenylmethyloxycarbonyl chemistry (Fmoc) by the MultiPep RSi Robot (Intavis). RBM peptide arrays were blocked in 5% BSA in 1X TBS-T for 2 hours, at room temperature. Arrays were overlaid with

1 μM ACE2-Fc recombinant protein (GenScript) [S1 Fig] overnight, at 4˚C. ACE2-Fc binding to RBM peptides was determined with an α-ACE2 (1:1000: Abcam–ab15348) primary antibody (4˚C, 4 hours), followed by an HRP-conjugated secondary antibody (1:5000: Sigma–A6154, room temperature, 1 hour) and finally ECL detection. All protein and antibody incubation steps were carried out under gentle agitation, and arrays washed following protein and antibody incubation steps in 1X TBS-T. Protein–peptide binding signal was assessed via observational changes in binding signal intensity (not peptide size/shape).

## Peptide synthesis and purification reagents

Fmoc-protected amino acids were purchased from CEM Corporation, and Pepceuticals. *N*, *N*-Dimethylformamide (DMF) and diethyl ether (Et$_2$O) were purchased from Rathburn. Triisopropylsilane (TIPS), Trifluoroacetic acid (TFA), *N*, *N*-Diisopropylethylamine (DIPEA), *N*,*N'*-diisopropylcarbodiimide (DIC), ethyl cyano (hydroxyimino) acetate (Oxyma Pure), fluorescein-5-isothiocyanate (FITC), Fmoc-Lys(Alloc)-OH, Tetrakis(triphenylphosphine)palladium (0) {Pd(PPh$_3$)$_4$)}, and phenylsilane were purchased from Fluorochem. Morpholine was purchased from Alfa Aesar. Dichloromethane (DCM) was purchased from VWR. Acetonitrile (MeCN) was purchased from Honeywell. TentaGel S RAM resin was purchased from Rapp Polymere. All other reagents were purchased from Sigma Aldrich.

## Solid phase peptide synthesis (SPPS) and peptide modifications

Peptides were synthesised on a 0.1 mmol scale using either a Biotage Initiator + Alstra (Biotage) or CEM Liberty Blue microwave assisted peptide synthesiser, with a TentaGel® S RAM resin (0.24 mmol/g). Coupling of Fmoc-protected amino acids (5 eq, 0.2 M in DMF) and unnatural/orthogonally protected amino acids (2.5 eq, 0.1 M in DMF) was achieved by treatment with DIC (5 eq, 0.5 M in DMF) and Oxyma Pure (5 eq, 0.5 M in DMF) at 90˚C for 2 minutes. β-branched amino acids and those following unnatural amino acids were double coupled. Deprotection was achieved by treatment with morpholine (20 % in DMF with 5 % formic acid, 4 mL) at 90˚C for 1 minute. The resin was washed with DMF between deprotection and coupling (4 x 4 mL), and after coupling (2 x 4 mL).

Peptides requiring *N*-terminal acetylation were treated on-resin with acetic anhydride (3 eq), DIPEA (4.5 eq) and DMF (7 mL for 0.1 mmol of resin) for 20 minutes with agitation. The resin was then washed with DMF (3 x 5 mL) and DCM (3 x 5 mL) prior to peptide cleavage and global deprotection.

Peptides containing Fmoc-Lys (Alloc)-OH were treated on-resin to selectively remove the Alloc protecting group. Pd (PPh$_3$)$_4$ (0.25 eq) and phenylsilane (25 eq) were pre-mixed in DCM (2 mL for 0.1 mmol of resin) and added to the resin, leaving for 4 h with agitation. The resin was then washed with DMF (2 x 5 mL) and DCM (2 x 5 mL) prior to peptide cleavage and global deprotection.

Peptides requiring a *C*-terminal fluorescent label were treated on-resin with FITC (2 eq), DIPEA (8 eq) and DMF (4 mL for 0.1 mmol of resin) following alloc deprotection of an orthogonally protected Lys. The resin was then washed with DMF (2 x 5 mL) and DCM (2 x 5 mL) prior to peptide cleavage and global deprotection.

## Peptide cleavage and global deprotection

Peptides were cleaved from the resin using a cocktail of TFA (95%), TIPS (2.5%) and H$_2$O (2.5%) for 3 hours with agitation. Peptides containing Cysteine (Cys) were cleaved from the resin using a cocktail containing TFA (94%), Ethane dithiol (EDT 2.5%), H$_2$O (2.5%) and TIPS (1%) for 3 hours with agitation. The resin was subsequently filtered and the TFA

evaporated using a stream of $N_2$, the peptide precipitated with cold $Et_2O$ and centrifuged (4500 rpm for 5 minutes). Peptides were dissolved in a mixture of $H_2O$ and MeCN with 0.1% TFA and lyophilized on a Christ Alpha 2–4 LO plus freeze dryer.

### Peptide purification [S2–S9 Figs]

Crude peptides were purified by reverse-phase high-performance liquid chromatography (RP-HPLC) using either an Agilent Technologies 1260 Infinity RP-HPLC system or a Dionex RP-HPLC system with Dionex P680 pumps and a Dionex UVD170U UV-vis detector, each with a Phenomenex Gemini column (5 mm C18, 250 x 21.2 mm). Purified peptides were analysed on a Shimadzu RP-HPLC system with Shimadzu LC-20AT pumps, a Shimadzu SIL20A autosampler and a Shimadzu SPD-20A UV-vis detector using a Phenomenex Aeris column (5 mm C18, 100 Å, 150 x 10 mm). Peptides were eluted with linear gradients at column-dependent flow rates (1 ml/min for the Aeris, 10 mL/min for the Gemini), where buffer A = 0.1% TFA in $H_2O$ and buffer B = 0.1% TFA in MeCN. Liquid chromatography mass spectrometry (LCMS) was performed on a Thermo Scientific LCQ Fleet Ion Trap Mass Spectrometer using positive mode electrospray ionisation ($ESI^+$). Where buffer A = 0.1% TFA and 5% MeCN in $H_2O$ and buffer B = 0.1% TFA and 5% $H_2O$ in MeCN, a linear gradient of 0–100% B over 20 min with a flow rate of 1 mL/min was used with a Reprosil-Gold column (3 mm C18, 150 x 4 mm).

### Fluorescent polarisation (FP)

FP assays were performed as described previously [16]. Briefly, direct binding of purified human ACE2-Fc (Q18 –S470) recombinant protein (GenScript) with fluorescently labelled RBM-derived peptides (FITC) was measured at an excitation/emission wavelength of 485/535nm using a Mithras LB 940 plate reader (Berthold Technologies). RBM-FITC peptides were incubated at a final concentration of 500 nM with increasing concentrations of ACE2 protein (0.012 μM– 6 μM) in PBS, and binding affinity ($K_D$) was calculated by nonlinear regression analysis of dose-response curves via GraphPad prism software (8.0).

### Culture and treatment of A549 cells

Human ACE2 stable expressing A549 Alveolar Type II Lung Epithelial cells were cultured as previously described [16]. A549 cells were pre-treated for 30 minutes with RBM-derived peptides at 5 μM (final concentration, diluted in media), followed by 0.1 μM (final concentration) incubation with $S_{RBD}$ -His protein (SARS-CoV-2 spike protein receptor binding domain, GenScript) for a further 3 hours.

### Western blot analysis

Protein lysates were harvested from A549 cells using 1X RIPA buffer (25 mM Tris–HCl, 150 mM NaCl, 0.5% sodium deoxycholate, 1% NP-40, 1 mM DTT, 0.1% SDS; pH 7.5), supplemented with protease and phosphatase inhibitors (Roche). SDS sample buffer (10% SDS, 300mM Tris-HCl, 0.05% bromothymol blue, 10% β-mercaptoethanol) was used to dilute protein samples. Samples were boiled for 10 minutes at 70˚C, and equal amounts of protein were loaded per well. Proteins were resolved on a 10% SDS-polyacrylamide gel, transferred to nitrocellulose membranes (GE Healthcare), stained with Ponceau-S (0.2% Ponceau-S red, 1% acetic acid) and blocked in 5% non-fat dry milk in 1X TBS-T (25mM, Tris-HCl; pH 7.6, 100 mM NaCl, 0.5% Tween-20).

Blocked membranes were incubated overnight at 4˚C with α-polyhistidine (1:3000 dilution, Sigma, H1029), α-ACE2 (1:1000 dilution, Abcam, ab15348) or α-α tubulin (1:1000, Sigma, T9026) primary antibodies. Membranes were then incubated in near-infrared (IRDye) secondary antibody (1:5000 dilution, 925–68072 or 925–32213, Li-Cor Biosciences) for 1 hour at room temperature. Membranes were washed in 1X TBS-T following each antibody step. Densitometry was carried out to measure fluorescent intensity of detected immunoreactive bands with Image Studio Lite (Licor Biosciences).

## Nano-luciferase fused ACE2 plasmid design

Nano-luciferase (nLuc) internalisation assays are a well-established and robust system for studying receptor internalisation [24, 25]. The internalisation assay was performed using a plasmid encoding for the human ACE2 enzyme fused with a nano-luciferase at the N-terminus. pcDNA5-nLuc-ACE2 was generated from three different plasmids, pΔSfiI-ΔRFP-SCRPSY-ACE2 [16], pcDNA5-FFA4-YFP [26], and pcDNA5-nLuc mGPR91 (generously gifted of Dr Brian Hudson). The ACE2 (ACE2-F-KpnI, pcDNA3-R) and the nano-Luciferase (pcDNA-MCS-F and NLUC-R-KpnI) sequences were amplified using Phusion enzyme kit (cat F5305, ThermoFisher) with the primers in Table 1. Subsequently, pcDNA5-FFA4-YFP was digested with HindIII and XhoI. The pcDNA5 backbone plasmid and the two PCR products were then ligated in a three-piece ligation reaction following the T4 ligation manufactory instructions.

## Culture and transfection of HEK293T

HEK293T cells were cultured in DMEM (Gibco) supplemented with 10% FBS (Gibco), 2mM L-glutamine (Gibco), 100U/I pen-strep (Gibco) and incubated at 5% $CO_2$, humified air, 37˚C. Transient transfections were performed with polyethyleneimine (PEI, Cat#23966 Polysciences Inc.). Briefly, nLuc-ACE2 plasmid (1.5 μg per 80% confluent petri dish) and 9 μg PEI were incubated for 10 minutes in 0.9% NaCl solution at room temperature, prior to being added to the petri dish dropwise. 24-hours post-transfection, cells were seeded into poly-d-lysine coated (ThermoFisher) 96 well white plates (CellStar TC).

## Luminescence internalisation assay

Transiently transfected HEK293T cells were washed and maintained in complete HBSS media (ThermoFisher) supplemented with 0.035% Sodium Bicarbonate (ThermoFisher) and 0.38% HEPES. HEK293T cells were then co-incubated for 2 hours with RBM-derived peptides (0.5, 5 or 50 μM, final concentrations) and 0.1 μM (final concentration) His-tagged SARS-CoV-2 Spike protein RBD (GenScript) or vehicle. Following 2-hour incubation period, HEK293T cells were further incubated for 10 minutes at 37˚C with the nLuc substrate Nanoglo (Promega) prior to measurement of total luminescence with the BMG Clariostar lab tech plate reader. The internalisation of membrane bound nLuc-ACE2, induced by $S_{RBD}$ protein, is indirectly detected by a reduction in luminescence signal.

**Table 1. Primers.**

| Primers | Sequence |
| --- | --- |
| pcDNA-MCS-F | TCCGGACTCTAGCGTTTAAACTTAAGCTT |
| NLUC-R-KpnI | TTTTTTGGTACCCGCCAGAATGCGTTCG |
| ACE2-F-KpnI | TTTTTTGGTACCTCAAGCTCTTCCTGGCTCCT |
| pcDNA3-R | CGAGCTCTAGCATTTAGGTGACACTATAG |

### SARS-CoV-2 spike protein–pseudovirus (PsV) neutralisation assay

As described previously [16], HEK293T cells stably expressing human ACE2 were seeded at $2x10^4$ cells into each well of a white opaque 96 well tissue culture plate (Perkin Elmer) and pre-treated for 1 hour with (i) DMSO only, (ii) RBM peptide(s) at 600 µM, 200 µM, 60 µM, 30 µM, 6 µM or 2 µM, or (iii) soluble human ACE2 (sACE2) at 1 µM [25 µL total well volume]. Following this, 25 µL of the SARS CoV-2 S-protein (Wuhan-Hu-1 strain, spike trimer) expressing PsV (containing a luciferase reporter) was then added [50 µL total well volume] and plates were incubated at 37˚C, 5% $CO_2$ for 48 hours. Resultingly, this halved (1:1 dilution) the final concentrations of RBM peptide / sACE2 treatments. Finally, luciferase activity (Indicative of S-mediated PsV entry) was measured and PsV entry inhibition (neutralisation) was calculated as a percentage of DMSO only negative controls.

### Cardiac microvascular endothelial cell culture and stimulation

Human coronary microvascular endothelial (CMEC) cells were cultured in endothelial growth medium MV2 (Promocell) until ~80% confluency. CMEC were serum starved 2 hours prior to any treatment(s) in DMEM containing 0.5% FBS. Following serum starving, cells were stimulated with 0.66 µg/mL (1 µg/$10^6$ cells) full-length recombinant SARS-CoV-2 S1 protein subunit (Cambridge Bioscience) for 5 hours. RBM peptides were pre-treated at 10 µM final concentration for 1 hour prior to S1 protein stimulation. Control cells were exposed to DMEM containing 0.5% FBS. Vehicle only control cells were pre-treated with 0.01% DMSO for 1 hour (same as RBM peptides). DMSO and RBM peptide treatments were diluted to appropriate concentrations in DMEM containing 0.5% FBS.

### Real-time reverse-transcription polymerase chain reaction

Total RNA was isolated using QIAzol Lysis Reagent (Qiagen) according to manufacturer's instructions and diluted in nuclease-free d$H_2O$ (Ambion/Life Technologies). cDNA was generated from total RNA using the High-Capacity cDNA Reverse Transcription Kits (Applied Biosytems). Real-time polymerase chain reaction (PCR) was performed via a 7900HT Fast Real-Time PCR System (Applied Biosystems), using the SyBR Green Master Mix (Applied Biosystems) and human primers specific to (i) GAPDH, (ii) IL-1β, (iii) IL-6, (iv) MCP-1 and (v) VCAM (Eurofins Genomics). Relative gene expression was calculated by the $2^{-\Delta\Delta Ct}$ cycle threshold method as previously described [27].

### Statistical analysis

Unless stated otherwise, all groups were analysed via a one-way ANOVA test. In the event a one-way ANOVA test measured statistical significance, follow up Dunnett's or Tukey's multiple comparison analysis was utilised to assess significance amongst groups. Where data was represented as MEAN ± SEM from N $\geq$ 3 independent experiments, results were determined significant by a $p$ value $< 0.05$. All statistical analyses were carried out using GraphPad Prism 8.0 software

## Results and discussion

### Generation of SARS-CoV-2 receptor binding motif peptides capable of selectively binding ACE2

Observation of the co-crystal structure of the SARS-CoV-2 Spike glycoprotein in complex with the ACE2 receptor clearly demonstrates that the vast majority of known contact residues

to ACE2 exist within the N437 –Y508 region of the viral protein, referred to as the receptor binding motif (RBM, corresponding to SARS-CoV-1 N424 –Y494) [5–8]. Previous analysis of the SARS-CoV-2 RBM–ACE2 PPI interface pinpoint the formation of hydrogen bonds between known contact residues, with distance between interactions proposed to be between 2.6–3.6 Å [5–8]. Strikingly, the 25mer peptides S438 –K462 (beginning of RBM) and E484 – Y508 (end of RBM) include the majority of these contact residues (5 and 10 respectively), with the exclusion of A475 (Fig 1). Moreover, S438 –K462 contains a Y451 –L–Y–R–L–F456 hexapeptide sequence (Fig 2), corresponding to the known SARS-CoV-1 RBM hexapeptide

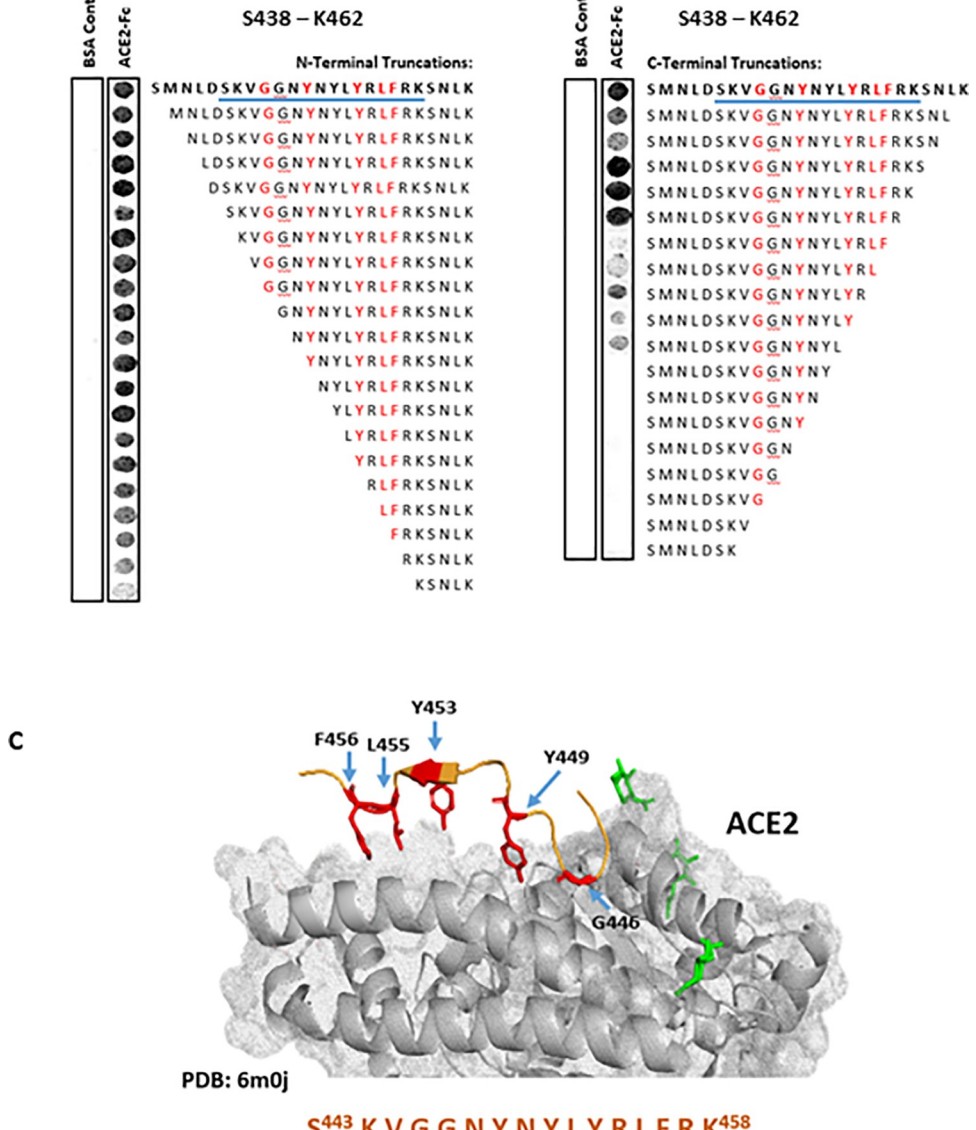

**Fig 2. Selecting a potential RBD–ACE2 PPI inhibitor peptide derived from SARS-CoV-2 RBM S438 –K462 peptide.** (A) N-terminal and (B) C-terminal stepwise truncation peptide array of RBM S438 –K462, overlaid with ACE2-Fc human purified protein. Underlined (blue) sequence represents shorter sequence peptide selected for assessment *in vitro*. (C) Structure of S443 –K458 SARS-CoV-2 RBM and the respective binding interface of ACE2 receptor; RBM interacting residues highlighted in red–adapted [7].

(Y438 –K–Y–R–Y–L443) shown to directly bind ACE2 and induce antiviral activity *in vitro* [19]. Resultingly, these two 25mer peptides were rationally selected for peptide array screening aimed at discovery novel decoy peptides of the RBD–ACE2 PPI.

Soluble recombinant ACE2 (Q18 –S740 truncate, S1 Fig) was found to bind to both S438 – K462 (Fig 2) and E484 –Y508 (Fig 3) peptides using a far-western technique, (as indicated by

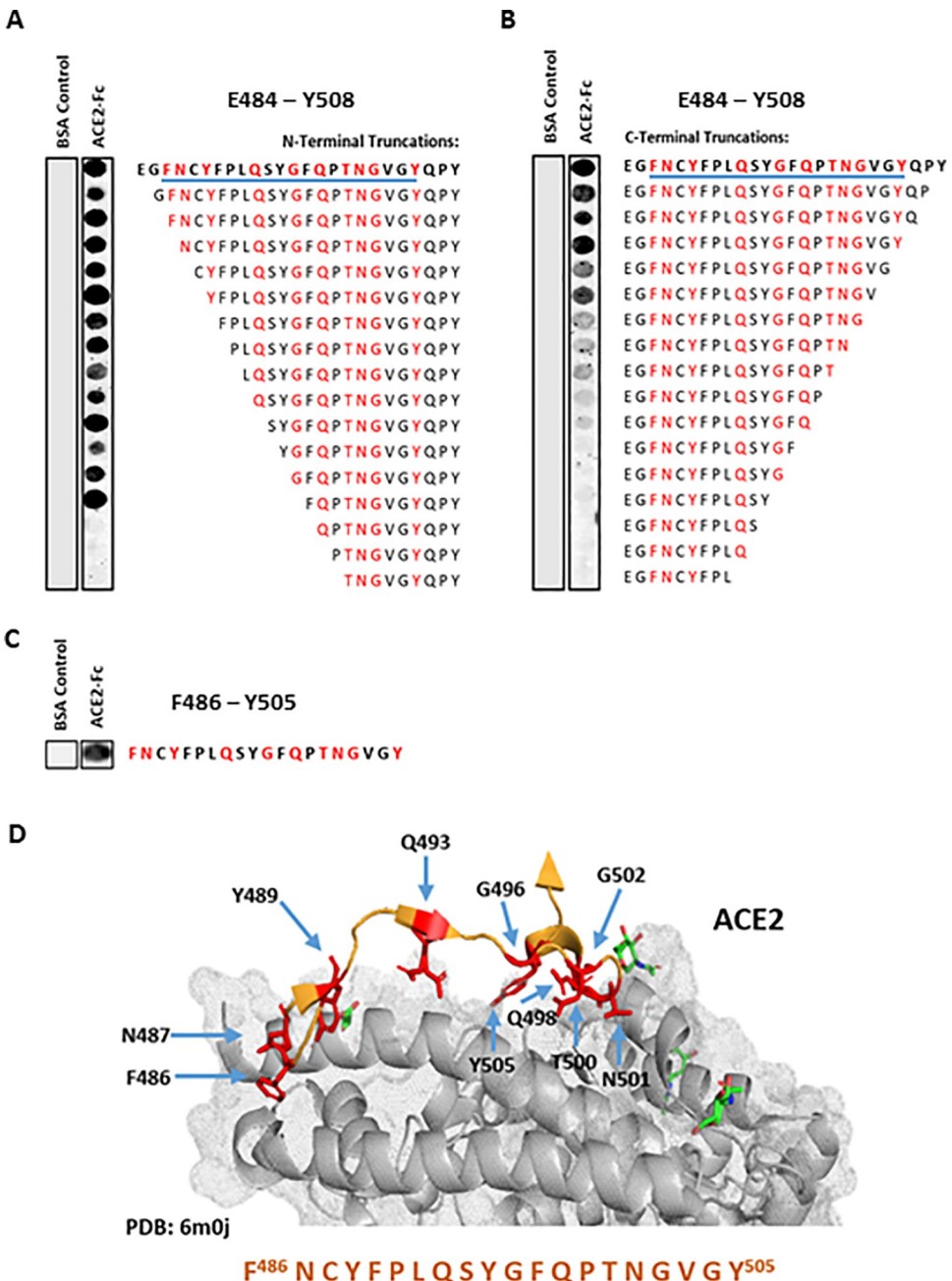

**Fig 3. Selecting a potential RBD–ACE2 PPI inhibitor peptide derived from SARS-CoV-2 RBM E484 –Y508.** (A) N-terminal and (B) C-terminal stepwise truncation peptide array of RBM E484 –Y508, overlaid with ACE2-Fc human purified protein. Underlined (blue) sequence represents shorter sequence peptide selected for assessment *in vitro*. (C) ACE2 binding to short sequence RBM 20mer; F486 –Y505. (D) Structure of F486 –Y505 SARS-CoV-2 RBM and the respective binding interface of ACE2 receptor; RBM interacting residues highlighted in red–adapted [7].

the dark binding spots). Serial truncation of the N- and C-terminus of peptide S438 –K462 (Fig 2) suggested that (i) removal of S438 –D442 resulted in little reduction in ACE2 binding signal vs. control peptide (Fig 2A), and (ii) removal of S459 –K462 led to a noticeable increase in ACE2 binding signal (i.e., observable increase in peptide spot intensity, Fig 2B). As no known ACE2 contact residues were contained within these truncated sequences (determined through observation of known co-crystal structures), the S443 –K458 16mer truncated peptide was selected for follow up assessment in biophysical and cellular assays–now referred to as receptor binding motif peptide 1 (RBM1) (Fig 2C). Truncation analysis of peptide E484 –Y508 (Fig 3) suggested that removal of E484, G485 and Q506 –Y508 resulted in no observable reduction in ACE2 binding vs. control peptide (Fig 3A and 3B). Although none of these residues are known direct ACE2 contact residues, E484 remains a residue of interest due to E484-specific point mutations (e.g., E484K) being found to increase SARS-CoV-2 infectivity [28]. Consequently, E484 –Y508 (RBM2A, 25mer) and F486 –Y505 (RBM2B, 20mer) peptides were selected for biophysical and cellular *in vitro* assessment (Fig 3C and 3D).

To confirm target engagement of RBM-derived peptides (Table 2) with ACE2, FITC-labelled peptides were co-incubated with increasing concentrations of purified recombinant human ACE2-Fc (Q18 –S740, peptidase domain) protein (final concentration range: 0.012 nM– 6 μM, Fig 4). All three peptides demonstrated direct binding to ACE2, with binding affinities (Kd) in the high nM to low μM range (RBM1 $K_d$ = 0.207 μM, RBM2A Kd = 1.13 μM, RBM2B Kd = 1.206 μM, N = 3, Fig 4A). No binding was observed with GST negative protein control, suggesting RBM peptides bound selectively to ACE2 (Fig 4B–4D). Although the RBM2A-Sc (RBM2A scrambled peptide negative control) appeared to bind to ACE2 in a dose-dependent manner (Fig 4C) saturation of ACE2 binding was not achieved and the data not interpreted via non-linear aggression analysis (GraphPad Prism 8.0). In total, the FP data suggested that we were successful in designing short sequence linear RBM-derived peptides capable of directly binding ACE2. Previous work using the peptide array approach has reliably generated highly-selective decoy peptides against pathological PPIs in a broad context of disease-indications, including heart failure (HSP20 –PDE4D) [29, 30], schizophrenia (DISC1 –FBWX7) [31], and cancer (c-Raf–PDE8A) [21, 32, 33]. All identified RBM peptides discovered in this study were tested as potential SARS-CoV-2 viral cell-entry inhibitors.

## RBM peptides do not prevent viral internalisation

To assess the ability of RBM-derived peptides to inhibit SARS-CoV-2 spike protein-mediated internalisation, the peptides were initially tested against the spike protein receptor binding

**Table 2. ACE2 interacting SARS-CoV-2 RBM peptides.** Peptides synthesised with and without a C-terminal FITC tag.

| Peptide Name (Residues, Length) | Sequence | Mass (Da) | | Purity (%) |
| --- | --- | --- | --- | --- |
| | | Calculated | Observed [M+H]$^+$ | |
| RBM1 [S443 –K458] | Ac-SKVGGNYNYLYRLFRK-NH$_2$ | 2019.34 | 2020.41 | 99 |
| RBM2A [E484-Y508] | Ac-EGFNCYFPLQSYGFQPTNGVGYQPY-NH$_2$ | 2918.19 | 2919.20 | 99 |
| | Ac-FNCYFPLQSYGFQPTNGVGY-NH$_2$ | 2343.60 | 2344.61 | 99 |
| RBM2A-Sc | Ac-NYGLQGSPFGYQETPYPFCNFVQYG-NH$_2$ | 2918.19 | 2919.25 | 96 |
| RBM1 (C-FITC) | Ac-SKVGGNYNYLYRLFRK-K(FITC)-NH$_2$ | 2537.34 | 2538.35 | 99 |
| RBM2A (C-FITC) | Ac-EGFNCYFPLQSYGFQPTNGVGYQPY-K(FITC)-NH$_2$ | 3433.43 | 3434.45 | 99 |
| RBM2B (C-FITC) | Ac-FNCYFPLQSYGFQPTNGVGY-K(FITC)-NH$_2$ | 2861.42 | 2862.43 | 99 |
| RBM2A-Sc (C-FITC) | Ac-NYGLQGSPFGYQETPYPFCNFVQYG-K(FITC)-NH$_2$ | 3433.43 | 3434.49 | 96 |

See S2–S9 Figs for HPLC traces. RBM, receptor binding motif; Sc, scrambled.

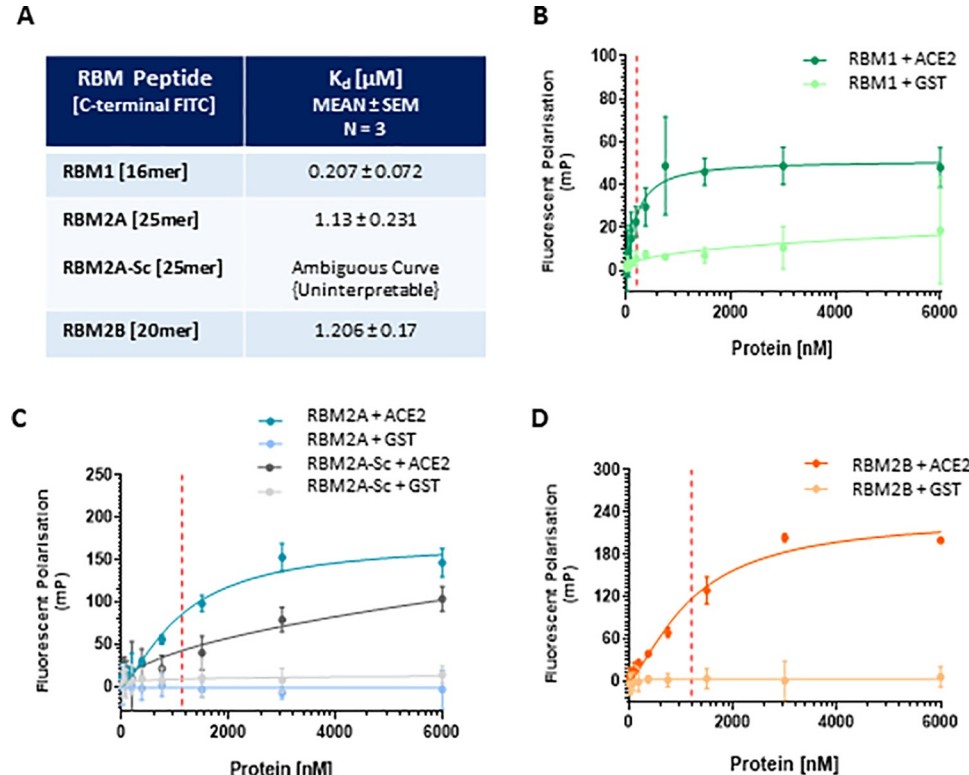

**Fig 4. Direct binding assay for ACE2 and RBM-derived peptides (FP; fluorescent polarisation).** Increasing concentrations of ACE2-Fc or GST purified recombinant protein (0.012 nM– 6 μM) were incubated with 500 nM of RBM peptide (FITC labelled) at room temperature, FP measured at 2 hours and binding affinities calculated via non-linear regression analysis (A). Binding saturation curves of (B) RBM1, (C) RBM2A and RBM2A-Sc, (D) RBM2B binding. GST represents negative protein control. Binding affinity (Kd, red vertical line) measurements represented as MEAN ± SEM, N = 3 independent experiments.

domain ($S_{RBD}$) in an assay utilising ACE2 overexpressing A549 lung epithelial cells (Fig 5A). RBM1 (56.98% ± 20.67%, Lane 8), RBM2A (44.89% ± 17.37%, Lane 6) and RBM2B (43.22% ± 21.35%, Lane 7) all induced a marked, yet non-significant reduction in $S_{RBD}$ mediated internalisation at 5 μM (Fig 5A). No significant difference was observed between RBM1, RBM2A and RBM2B. RBM-Sc did not significantly inhibit $S_{RBD}$-mediated internalisation (94.19% ± 30.35%, Fig 5A, Lane 5). In an attempt to cover a larger segment of the PPI interface, a combination of RBM1 and RBM2B peptides was also tested, however, this did not potentiate the inhibitory effects observed with respective monotherapies (89.53% ± 47.23%, Fig 5A, Lane 9). Whether these peptides had an antagonistic effect on one another was not assessed.

RBM peptides were then tested against $S_{RBD}$ mediated ACE2 internalisation using a luciferase reporter ACE2 overexpressing HEK293T cell line (Fig 5B(i)). Recombinant $S_{RBD}$ (100 nM) induced internalisation resulting in a significant reduction in luminescence signal (Fig 5B(ii), *** P < 0.001) observed on the cell surface. RBM2A and RBM2B, but not RBM1, appeared to partially rescue $S_{RBD}$-mediated internalisation at 50 μM (not seen at 0.5 or 5 μM), as seen by a non-significant reduction in luminescence signal observed between vehicle treated and $S_{RBD}$ treated conditions (Fig 5B, RBM2A: Blue–Lane 3, RBM2B: Orange–Lane 3). No significant difference was observed between RBM peptides.

It is worth noting that, even though (i) cells were pre-treated with RBM peptides prior to the application of $S_{RBD}$ (giving the peptides time to bind ACE2 without direct competition

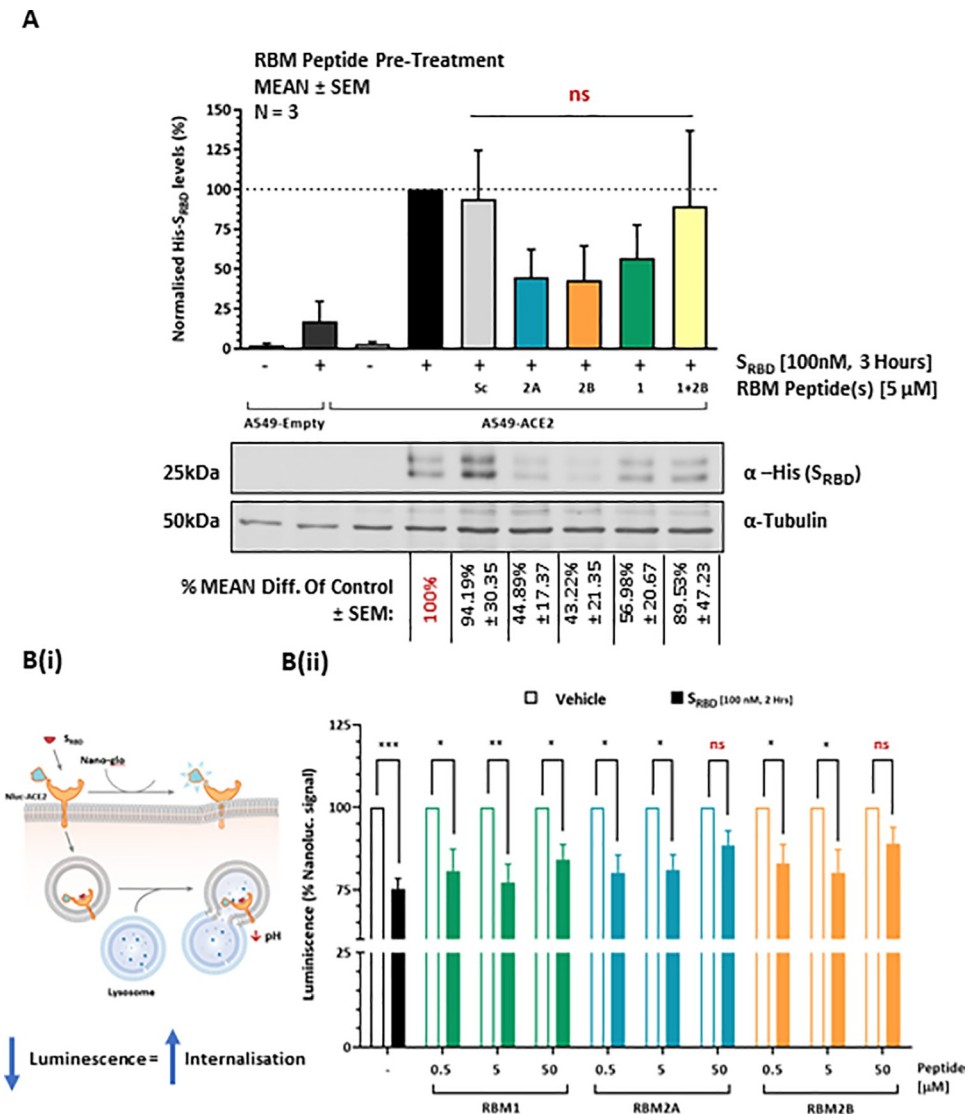

**Fig 5. $S_{RBD}$ internalisation in human ACE2 overexpressing human cell lines.** (A) western immunoblot detection of internalised $S_{RBD}$ protein in ACE2 overexpressing (stable) A549 cells following 30 minutes pre-treatment with (i) vehicle only (lane 4), (ii) RBM2A-Sc (lane 5), (iii) RBM2A (lane 6), (iv) RBM2B (lane 7), (v) RBM1 (lane 8), RBM1 + RBM2B (lane 9). (B) Changes to $S_{RBD}$ mediated internalisation in transiently overexpressing ACE2 (luciferase) HEK293T cells were measured following co-treatment with $S_{RBD}$ (100 nM) and (i) RBM1, (ii) RBM2A, (iii) RBM2B peptides (0.5, 5 and 50 μM). Levels of internalised $S_{RBD}$ measured as % difference of vehicle only control. All data represented as MEAN ± SEM, N = 3 independent experiments. * P < 0.05, ** P < 0.01, *** P < 0.001, n.s. not significant.

with $S_{RBD}$), and (ii) the concentration of RBM peptides was 5- to 500-fold higher than $S_{RBD}$ (100 nM), RBM peptides (alone or in combination) were unable to significantly abolish $S_{RBD}$ internalisation. These findings suggest that RBM1-2B are unable to effectively outcompete $S_{RBD}$ protein binding to ACE2.

Next, we attempted to evaluate the RBM peptides against the full-length spike trimer protein (Wuhan-Hu-1 strain) in an ACE2 overexpressing HEK293 pseudovirus (PsV) neutralisation assay (Fig 6). As expected, and previously demonstrated [16], soluble ACE2 protein completely blocked spike PsV cell entry at 0.5 μM. However, pre-treatment with RBM1, RBM2B or combined RBM1 and RBM2B did not inhibit PsV cell entry, even at a

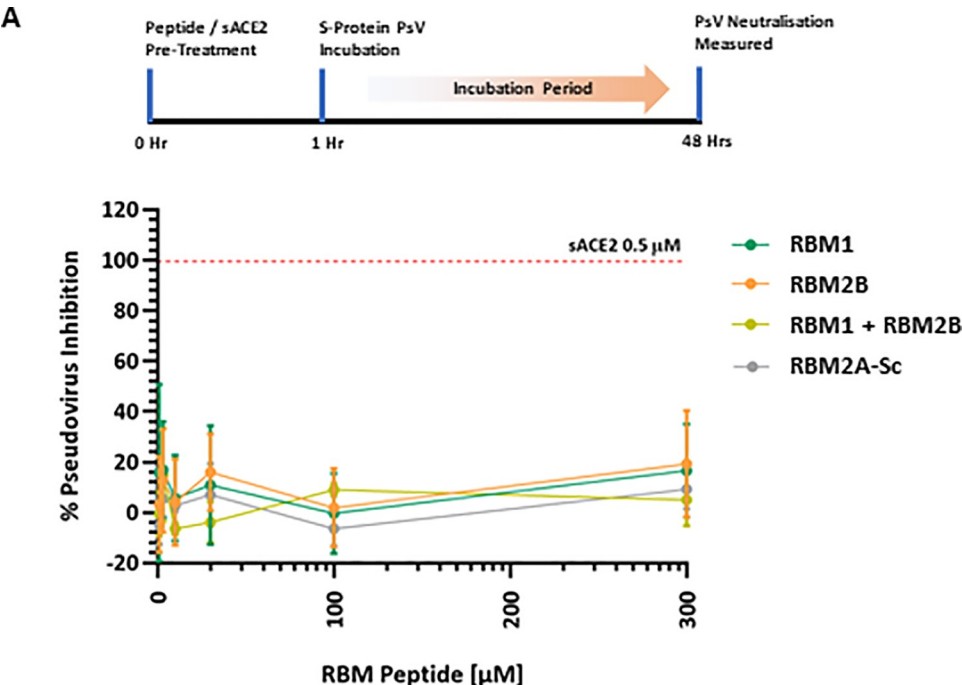

**Fig 6. Spike protein (full-length trimer, Wuhan-Hu-1 strain) pseudovirus neutralisation assay in human ACE2 overexpressing HEK293T cells.** RBM peptides were pre-treated (1 hour) at 300 µM, 100 µM, 30 µM, 10 µM, 3 µM and 1 µM, followed by 48-hour co-incubation with spike protein PsV (luciferase labelled) and subsequent luminescence detection. sACE2 (human ACE2 protein) represents positive PsV neutralisation control. Data represented as MEAN % difference of DMSO only negative control (100%) ± SEM, N = 3 independent experiments. *PsV*, *Pseudovirus*.

concentration of 300 µM. These data suggest that the spike trimer can outcompete RBM-derived peptides for binding to ACE2, resulting in a lack of inhibitory activity.

Our findings, outlined above, strengthen the hypothesis that linear short sequence native RBM1-2B peptides represent a suboptimal approach to inhibiting ACE2-mediated SARS-CoV-2 host cell entry. Whether rational chemical modifications to the current structure of RBM1-2B peptides would significantly improve ACE2 binding affinity (and subsequent anti-viral activity), as seen with double stapled α1 helix ACE2-derived peptides vs. their respective linear analogues [18], remains unknown. It is worth noting that RBM peptides lack helical properties and are therefore not suitable candidates for peptide stapling. As such, other approaches (e.g., peptide cyclisation) should be considered as a potential way of improving RBM-peptides structure-activity-relationship. Furthermore, analysis of global genome sequencing datasets detailing identified mutations within the receptor binding domain have identified variants that possess higher ACE2 binding affinity (e.g., Y453F and N501Y) compared with the original Wuhan-1 strain [34, 35]. Resultingly, these mutations could be leveraged in future efforts aimed at selecting peptide sequences that possess higher ACE2 binding affinity. However, with picomolar SARS-CoV-2 'miniproteins' demonstrating enhanced molecular interactions with ACE2 and potent anti-viral activity against SARS-CoV-2 [36], our data (and others like it [16–18]) further indicate that larger molecules with significantly higher ACE2 binding affinity represent a more apt approach to competitively blocking SARS-CoV-2 cell entry.

## RBM peptides do not inhibit S1 protein induced inflammation

Finally, to determine whether our linear RBM peptides had the ability to perturb SARS-CoV-2 mediated inflammation (directly associated with the cytokine storm [37]) we tested them in a

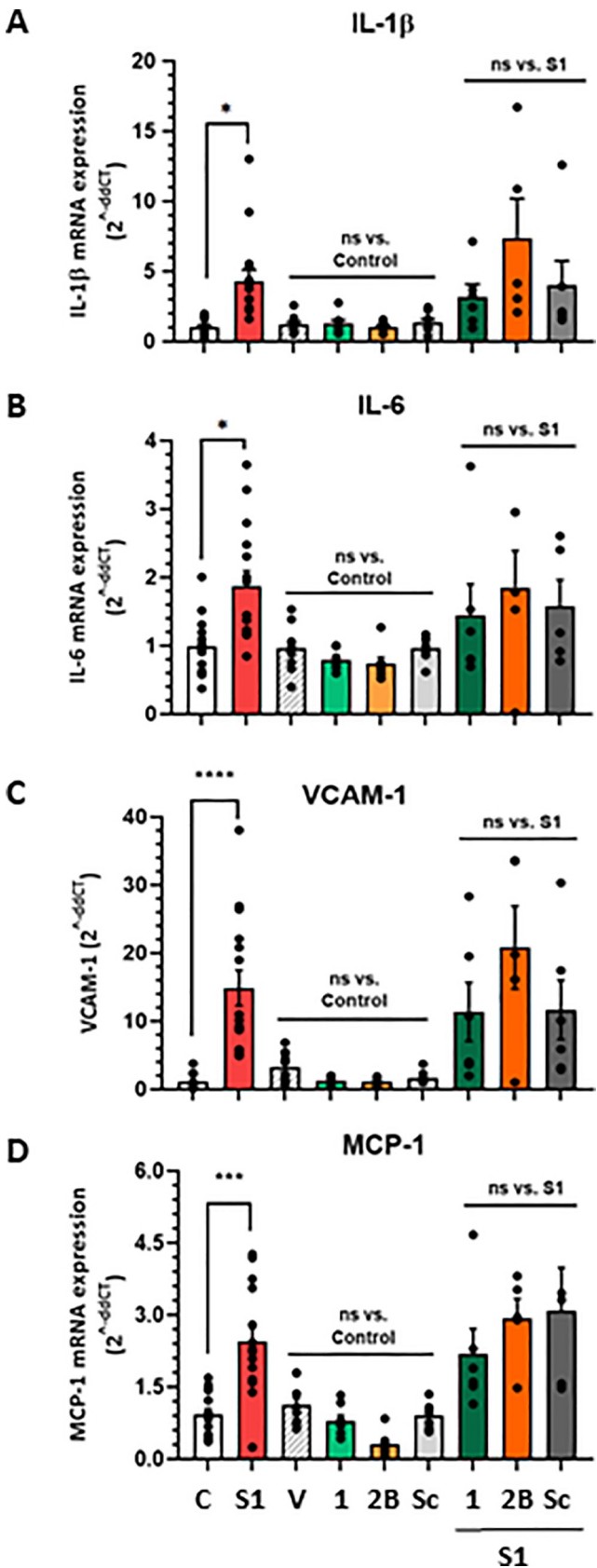

**Fig 7. Effect of RBM peptides on recombinant S1 protein-induced inflammation in ACE2 endogenously expressing human coronary microvascular endothelial cells.** Total RNA extracted from CMEC and gene expression of (A) IL-1β, (B) IL-6, (C) VCAM-1, (D) MCP-1 pro-inflammatory markers determined via RT-PCR (normalised to GAPDH). RBM peptides treated alone (10 μM) or pre-treated (10 μM, 1 hour) following S1 protein 5-hour incubation (9 nM). Data represented as MEAN ± SEM, N ≥ 3 independent experiments. * P < 0.05, *** P < 0.001, **** P < 0.0001, n.s. not significant. *C, Control; S1, spike protein S1 subunit; V, vehicle only control; 1, RBM1 peptide; 2B, RBM2B peptide; Sc, RBM2A-Scrambled peptide.*

physiologically relevant human epithelial (CMEC) cell line that endogenously expresses ACE2. In these cells, S1 protein significantly upregulated mRNA expression of four pro-inflammatory markers associated with SARS-CoV-2 induced cytokine storm: IL-1β (Fig 7A, * P < 0.05, lanes 1–2), IL-6 (Fig 7B, * P < 0.05, lanes 1–2), VCAM (Fig 7C, **** P < 0.0001, lanes 1–2) and MCP-1 (Fig 7D, *** P < 0.001, lanes 1–2). RBM1 and RBM2B treatment (10 μM) did not mimic S1 protein-induced inflammation, indicating RBM peptides are not pro-inflammatory in this context (Fig 7A–7D, ns–not significant, lanes 3–6). Additionally, pre-treatment of cells with RBM1 or RBM2 did not attenuate the S1 induced inflammatory response (Fig 7, ns–not significant, lanes 7–9). In line with our other cellular assays (Figs 5 and 6), RBM peptide treatment concentrations (10 μM) were >100-fold higher than S1 protein (9 nM), reinforcing the notion that RBM peptides cannot effectively compete with S1 for binding to ACE2.

In summary, our work builds on the existing hypothesis that short linear peptide sequences derived from the native SARS-CoV-2 RBD–ACE2 PPI interface, represent a suboptimal therapeutic approach to blocking viral internalisation. Larger ligands (e.g., soluble ACE2, miniproteins, antibodies, nanobodies) that possess higher binding affinity for ACE2, robust stability and significantly enhanced secondary structure(s) are likely to confer superior antiviral efficacy, and future research should consider this when developing associated SARS-CoV-2 cell entry inhibitor therapeutics.

## Supporting information

**S1 Fig.** Human purified recombinant ACE2-Fc (Q18-S740; GenScript), detected via SDS-PAGE by (left) Coomassie staining and (right) western immunoblotting with an αACE2 rabbit polyclonal antibody. Lane 1: Ladder, Lane 2: ACE2-Fc protein. ACE2-Fc protein detected at expected molecular weight (~110.5kDa).
(TIF)

**S2 Fig. HPLC traces of RBM1 (S438-K462) on 2 gradients.**
(TIF)

**S3 Fig. HPLC traces of RBM2A (E484-Y508) on 2 gradients.**
(TIF)

**S4 Fig. HPLC traces of RBM2B (F486-Y505) on 2 gradients.**
(TIF)

**S5 Fig. HPLC traces of RBM2A-Scrambled on 2 gradients.**
(TIF)

**S6 Fig. HPLC traces of RBM1 (S438-K462)-FITC on 2 gradients.**
(TIF)

**S7 Fig. HPLC traces of RBM2A (E484-Y508)-FITC on 2 gradients.**
(TIF)

**S8 Fig. HPLC traces of RBM2B (F486-Y505)-FITC on 2 gradients.**
(TIF)

**S9 Fig. HPLC traces of RBM2A-scrambled-FITC on 2 gradients.**
(TIF)

**S1 Raw image.**
(PDF)

**S1 Data.**
(XLSX)

**S2 Data.**
(XLSX)

**S3 Data.**
(XLSX)

**S1 Dataset.**
(PZFX)

**S2 Dataset.**
(PZFX)

## Author Contributions

**Formal analysis:** Amit Mahindra, Gonzalo Tejeda, Mario Rossi, Omar Janha, Imogen Herbert, Caroline Morris, Danielle C. Morgan, Wendy Beattie, Augusto C. Montezano, Brian Hudson, Connor M. Blair.

**Funding acquisition:** Connor M. Blair.

**Investigation:** Amit Mahindra, Gonzalo Tejeda, Mario Rossi, Omar Janha, Imogen Herbert, Caroline Morris, Danielle C. Morgan, Wendy Beattie, Augusto C. Montezano, Brian Hudson, Connor M. Blair.

**Methodology:** Amit Mahindra, Gonzalo Tejeda, Mario Rossi, Omar Janha, Imogen Herbert, Caroline Morris, Danielle C. Morgan, Wendy Beattie, Augusto C. Montezano, Brian Hudson.

**Project administration:** Amit Mahindra, Connor M. Blair.

**Supervision:** Andrew B. Tobin, David Bhella, Rhian M. Touyz, Andrew G. Jamieson, George S. Baillie, Connor M. Blair.

**Writing – original draft:** Connor M. Blair.

**Writing – review & editing:** Amit Mahindra, Gonzalo Tejeda, Mario Rossi, Caroline Morris, Danielle C. Morgan, Andrew B. Tobin, David Bhella, Rhian M. Touyz, Andrew G. Jamieson, George S. Baillie, Connor M. Blair.

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
