## [Decision Letter · Decision Letter 0]

22 Sep 2021

PONE-D-21-28336Peptides derived from the SARS-CoV-2 Receptor Binding Motif  bind to ACE2 but do not block ACE2-mediated host cell entry or pro-inflammatory cytokine induction.PLOS ONE

Dear Dr. Blair,

Thank you for submitting your manuscript to PLOS ONE. After careful consideration, we feel that it has merit but does not fully meet PLOS ONE’s publication criteria as it currently stands. Therefore, we invite you to submit a revised version of the manuscript that addresses the points raised during the review process.

We look forward to receiving your revised manuscript.

Kind regards,

Tianwen Wang, Ph.D.

Academic Editor

PLOS ONE

Journal Requirements:

"CMB, MR, GT and OJ are supported through the WT ISSF COVID Response Award (204820/Z/16/Z). AM is supported through the EPSRC (Research Project Grant EP/N034260/2) and CM is supported through Dstl (DSTLX-1000141308). DCM thanks the EPSRC for a studentship (EP/N509668/1 and EP/R513222/1). RMT is supported through a BHF Chair award (CH/4/29762). ACM is supported by a University of Glasgow Walton Fellowship. DB and IH were supported by the Medical Research Council (MC UU 12014/7)."

"Acknowledgements:

CMB, MR, GT and OJ are supported through the WT ISSF COVID Response Award (204820/Z/16/Z). 

AM is supported through the EPSRC (Research Project Grant EP/N034260/2) and CM is supported through Dstl (DSTLX-1000141308). DCM thanks the EPSRC for a studentship (EP/N509668/1 and EP/R513222/1). RMT is supported through a BHF Chair award (CH/4/29762). ACM is supported by a University of Glasgow Walton Fellowship. DB and IH were supported by the Medical Research Council (MC UU 12014/7).

Author Contributions:

CMB and AM designed the study. CMB, AM, GT, MR, OJ, IH, DCM, CM, WB and ACM performed the experiments. CMB, AM and GSB wrote the manuscript. ACM, BH, ABT, DB, RMT, AGJ and GSB provided critical support and supervision of study. All authors read and approved manuscript."

6. If tables are embedded in the manuscript and ALSO loaded as separate files, please delete the separate files, leaving only the tables in the manuscript file.  If they have uploaded the files as figure files and they are not in the manuscript, use the sendback.

Reviewers' comments:

Reviewer's Responses to Questions

**Comments to the Author**

1. Is the manuscript technically sound, and do the data support the conclusions?

Reviewer #1: Partly

Reviewer #2: Yes

2. Has the statistical analysis been performed appropriately and rigorously? 

Reviewer #1: Yes

Reviewer #2: Yes

3. Have the authors made all data underlying the findings in their manuscript fully available?

Reviewer #1: Yes

Reviewer #2: Yes

4. Is the manuscript presented in an intelligible fashion and written in standard English?

Reviewer #1: Yes

Reviewer #2: Yes

5. Review Comments to the Author

Reviewer #1: This study aims to evaluate whether the peptides derived from the SARS-CoV-2 RBM would be capable to block ACE2-mediated host cell entry. Although through a serial of experimental testing none of the peptides can fulfill the mission, these findings reinforce the fact that larger biologics may better suited as SARS-CoV-2 cell entry inhibitors. As COVID19 coronal virus pandemic represents a serious threat to the public heath globally, the efforts this research group put into this study are much appreciated.

My major issue lies in the affinity base line (line 81-85) been set for screening for peptide binding candidates is not optimal. This study is forcing on studying SARS-CoV-2, therefore using the affinity value from SARS-CoV-1 RBM study is not appropriate. Especially when it’s been shown that SARS-CoV-2 binds ACE2 at least 10 times more tightly than the corresponding spike protein of SARS-CoV-1 (Wrapp et al, 2020). That could contribute partially to the peptides’ ultimate failure in blocking entry. There’s an urgency of SARS-CoV-2 research and in the meantime, the knowledge regarding SARS-CoV-2 has been frequently updating, please make sure that the “introduction” section presents the most up-to-date and accurate information.

The end goal of this study is clear, however, the rationale behind selecting which region on SARS-Co-2 RBM is not clearly stated. Instead of only saying “through observation…..rationally selected” (line 267-269), a more detailed analysis is required to list how and why those ones have been picked. Do those residues have polar interaction with ACE2? Hydrogen-bonding, slat bridge or… What is the distance between the atoms in Å? Those can be analyzed through PyMOL or other software the author would prefer to use. The additional analysis can be reflected in Figure 1. If other group has investigated the interactions, the results need to be cited properly and clearly stated as well. This is the base and core of the study, without data backup and detailed analysis the significance of the whole study is diluted.

A control of a positive SARS-CoV-1 cell-entry inhibitor should be included as an experimental control in the Luminescence Internalization Assay. With adding an inhibitor, there should not be much or any of decreasing in luminescence signal. At least to show the experimental set up is sound. Otherwise, it’s less convincing by showing a positive action (internalization) by a negative phenotype (luminescence). I would strongly suggest the author to consider conjugating the target proteins with pH sensitive fluorescence probe that only gets activated under lower pH. Then, internalization will correspond with higher fluorescence signals.

Reviewer #2: The manuscript by Amit et al presented negative results. they reported that the native RBM peptide is unable to inhibit the spike-ACE2 interaction and thus suggesting alternative therapies. The manuscript is written well, the data is well organized and can be accepted after minor revision.

1. please mention the peptide sequence in the abstract.

2. data about the variants should be added at least alignment of this peptide in the the reference and variants so that another insights can be viewed for more work.

6. PLOS authors have the option to publish the peer review history of their article (what does this mean?). If published, this will include your full peer review and any attached files.

Reviewer #1: No

Reviewer #2: **Yes: **Abbas Khan

---

## [Author Response · Author response to Decision Letter 0]

1 Oct 2021

I have included in the 'response to reviewers.docx' file that was uploaded. However, a copy of the response is below:

Response to Reviewer 1:

a. Regarding comments RE: Lines 81-85… 

I have made amendments to acknowledge the fact that SARS-CoV-2 spike protein binds ACE2 with significantly higher affinity than SARS-CoV-1 spike protein. I have also acknowledged that the binding affinity of our peptides with ACE2 is needed to be significantly higher than the SAR-CoV-1 RBM Hexapeptide if to be competitive against the spike protein. (Wrapp, et al 2020) reference has also been added in. I hope this will suffice:

“Previous work utilising a SARS-CoV-1 spike RBM derived hexapeptide (Y438-K-Y-R-Y-L443) demonstrated a modest ability to directly bind ACE2 (Kd = 46 µM) and inhibit coronavirus NL63 cell entry and subsequent viral replication 19. Although this peptide was not tested against SARS-CoV-1 viral entry, these findings highlight the possibility for a SARS-CoV-2 RBM-derived peptide to represent a potentially promising approach to the development of novel antiviral decoy peptides. However, as the spike protein corresponding to SARS-CoV-2 has been shown to bind ACE2 10 times more tightly than SARS-CoV-1, the binding affinity of SARS-CoV-2 derived peptide(s) will need to be significantly higher in order to be competitive against the spike protein 33”

b. Regarding comments RE: rational behind selecting which region on SARS…

I have added further explanation as to why we selected these two peptides as high confidence starting points for the development of RBD – ACE2 PPI decoy peptides (Lines 275-290). Selection is initially derived from data that exists within the public domain – i.e. existing solved co-crystal structures and related publications that have already carried out extensive analysis. I have referenced these in text appropriately.

As analysis regarding contact residues between SARS-CoV-2 RBM and ACE2 has already been extensively investigated and published (references already included in manuscript), I have not included data in Figure 1 that details my own analysis as it would be directly replicating data that is already available and therefore add no novel value to the figure/study. As such, Figure 1 remains the same – highlighting the known contact residues of SARS-CoV-2 RBM that directly interact with ACE2. I have also highlighted these residues (in red) in a published (solved) co-crystal structure of the SARS-CoV-2 RBD – ACE2 protein-protein interaction for qualitative purposes (visualisation). This structure can be found on PDB and the protein structure I have used in the figure was generated using PyMol software. I hope this suffices.

c. Regarding comments RE: showing luminescence experimental set up is sound…

Having discussed this with Dr Brian Hudson and Dr Mario Rossie (the two researchers responsible for developing and carrying out the luciferase assay), they have forwarded the following response to your comment(s): 

‘We appreciate the reviewers’ comment, however we would also like to highlight that these internalization approaches based on the pH dependence of the NanoLuc activity have already been tested extensively by the scientific community. These tools have in fact been considered well established, valuable approaches for quantification of cell membrane protein internalization, especially in HEK293 cell lines. In the resubmitted version of the paper, you will find two new reference articles showing nano luciferase as a sensitive bioluminescent reporter to measure quantitatively the internalization of cell membrane receptors.

“Quantitative measurement of cell membrane receptor internalization by the nanoluciferase reporter: Using the G protein-coupled receptor RXFP3 as model. Y. Liu et al. / Biochimica et Biophysica Acta 1848 (2015) 688–694”, “A luminescent assay for real-time measurements of receptor endocytosis in living cells. M.B. Robers et al. / Analytical Biochemistry 489 (2015) 1e82“’

I hope this amendment/response will suffice.

Response to Reviewer 2:

a. Sequences added to abstract as requested.

b. Although we did not generate any data regarding point mutations within our peptides that would confer higher ACE2 binding affinity, I have included a small section in the updated tracked changed document (Lines 406-410) briefly discussing the possibility of selecting for already known RBD point mutations that could lead to higher ACE2 binding affinity of our selected RBM peptides. Due to grant funding/duration coming to an end, we are unable to carry out further experiments relating to mutations. I hope this statement + relevant addition of references will suffice.

“Furthermore, analysis of global genome sequencing datasets detailing identified mutations within the receptor binding domain have identified variants that possess higher ACE2 binding affinity (e.g. Y453F and N501Y) compared with the original Wuhan-1 strain 33,34. Resultingly, these mutations could be leveraged in future efforts aimed at selecting peptide sequences that possess higher ACE2 binding affinity.”

---

## [Decision Letter · Decision Letter 1]

11 Oct 2021

PONE-D-21-28336R1Peptides derived from the SARS-CoV-2 Receptor Binding Motif  bind to ACE2 but do not block ACE2-mediated host cell entry or pro-inflammatory cytokine induction.PLOS ONE

Dear Dr. Blair,

Thank you for submitting your manuscript to PLOS ONE. After careful consideration, we feel that it has merit but does not fully meet PLOS ONE’s publication criteria as it currently stands. Therefore, we invite you to submit a revised version of the manuscript that addresses the points raised during the review process.

We look forward to receiving your revised manuscript.

Kind regards,

Tianwen Wang, Ph.D.

Academic Editor

PLOS ONE

Journal Requirements:

Reviewers' comments:

Reviewer's Responses to Questions

**Comments to the Author**

1. If the authors have adequately addressed your comments raised in a previous round of review and you feel that this manuscript is now acceptable for publication, you may indicate that here to bypass the “Comments to the Author” section, enter your conflict of interest statement in the “Confidential to Editor” section, and submit your "Accept" recommendation.

Reviewer #1: All comments have been addressed

Reviewer #2: All comments have been addressed

2. Is the manuscript technically sound, and do the data support the conclusions?

Reviewer #1: Yes

Reviewer #2: Yes

3. Has the statistical analysis been performed appropriately and rigorously? 

Reviewer #1: Yes

Reviewer #2: I Don't Know

4. Have the authors made all data underlying the findings in their manuscript fully available?

Reviewer #1: Yes

Reviewer #2: Yes

5. Is the manuscript presented in an intelligible fashion and written in standard English?

Reviewer #1: Yes

Reviewer #2: Yes

6. Review Comments to the Author

Reviewer #1: Although no additional experiments were performed, the author did rationalized pretty well.

The newly inserted references must be listed in the same format as the rest and properly aligned.

Reviewer #2: thank for revising your manuscript.

7. PLOS authors have the option to publish the peer review history of their article (what does this mean?). If published, this will include your full peer review and any attached files.

Reviewer #1: No

Reviewer #2: No

---

## [Author Response · Author response to Decision Letter 1]

4 Nov 2021

PLOS One Rebuttal Letter

PONE-D-21-28336

Peptides derived from the SARS-CoV-2 Receptor Binding Motif bind to ACE2 but do not block ACE2-mediated host cell entry or pro-inflammatory cytokine induction.

Corresponding Author: Dr. Connor M. Blair (connor.blair@glasgow.ac.uk)

(0) References have been updated so that they run in an appropriate numerical/sequential order in text and in reference list – as requested. None of the references included have been retracted. However, reference 37 (Yang, et al. 2021 Science) has had a ‘correction’ published: {correction: Yang L, Xie X, Tu Z, Fu J, Xu D, Zhou Y. The signal pathways and treatment of cytokine storm in COVID-19. Signal Transduct Target Ther. 2021;6(1):326}. Correction was published Aug 31st 2021 ( ~7 weeks after original publication). Original publication has been updated on Nature Publishing Groups webpage and therefore the original citation marked reference 37 is an up-to-date draft and will direct you to the intended/appropriate review article. I hope this is appropriate.

(1) Figures have been re-uploaded following analysis on PACE.

(2) Manuscript Style Requirements: submitted manuscript is .docx file, Calibri (body), double spaced, with abbreviations stated early on in text. Vancouver style was used for referencing. Each figure has been attached as a separate/individual file. Please let me know specifically if there is any other aspect of style in which I am not meeting standards.

(3) Amended Statements:

a. Funding Statement should read: 

‘CMB, MR, GT and OJ are supported through the WT ISSF COVID Response Award (204820/Z/16/Z). AM is supported through the EPSRC (Research Project Grant EP/N034260/2) and CM is supported through Dstl (DSTLX-1000141308). DCM thanks the EPSRC for a studentship (EP/N509668/1 and EP/R513222/1). RMT is supported through a BHF Chair award (CH/4/29762). ACM is supported by a University of Glasgow Walton Fellowship. DB and IH were supported by the Medical Research Council (MC UU 12014/7).’

(4) & (5) Minimal data files have been updated in the supplementary files, including:

a. Minimal data file relating to fluorescent polarisation experiments: 

name: ‘Fig 4_Fluorescent Polarisation_Min Data.xlsx’

b. Minimal data file relating to western blotting experiments (including uncropped IBs):

name: ‘Fig 5A_WB Internalisation Assay_Min Data.pzfx’ AND ‘Fig 5A_WB Internalisation Assay_ Uncropped WBs.pdf’

c. Minimal data file relating to luminescence experiments:

name: ‘Fig 5B_N-Luc Internalisation Assay_Min Data.pzfx’

d. Minimal data file relating to pseudovirus experiments:

name: ‘Fig 6_Spike PsV Inhibition Assay_Data.xlsx’

e. Minimal data file relating to inflammatory marker experiments:

name: ‘Fig 7_CMEC Inflammation Marker_Min Data.xlsx’

(6) Tables in manuscript: I have not included table 2 within the original main body of the manuscript submitted. Instead, table 2 was found within the combined figure file that was attached/submitted. As per request, I have removed Table 2 from the original combined Figure File and have now inserted into the main body of the manuscript at the appropriate point. Thus, both Table 1 and 2 are within main body of manuscript only. 

The table that is included in Figure 1 is part of Figure 1 and I therefore request that it remains this way. I hope this meets the requirements. 

Note that all figures (including supplementary) have been uploaded as separate .TIF files as per request. I have also included a PDF file with ALL figures. Please select whichever comes out with highest resolution.

(7) Response to Reviewer 1:

a. Regarding comments RE: Lines 81-85… 

I have made amendments to acknowledge the fact that SARS-CoV-2 spike protein binds ACE2 with significantly higher affinity than SARS-CoV-1 spike protein. I have also acknowledged that the binding affinity of our peptides with ACE2 is needed to be significantly higher than the SAR-CoV-1 RBM Hexapeptide if to be competitive against the spike protein. (Wrapp, et al 2020) reference has also been added in. I hope this will suffice:

“Previous work utilising a SARS-CoV-1 spike RBM derived hexapeptide (Y438-K-Y-R-Y-L443) demonstrated a modest ability to directly bind ACE2 (Kd = 46 µM) and inhibit coronavirus NL63 cell entry and subsequent viral replication 19. Although this peptide was not tested against SARS-CoV-1 viral entry, these findings highlight the possibility for a SARS-CoV-2 RBM-derived peptide to represent a potentially promising approach to the development of novel antiviral decoy peptides. However, as the spike protein corresponding to SARS-CoV-2 has been shown to bind ACE2 10 times more tightly than SARS-CoV-1, the binding affinity of SARS-CoV-2 derived peptide(s) will need to be significantly higher in order to be competitive against the spike protein 33”

b. Regarding comments RE: rational behind selecting which region on SARS…

I have added further explanation as to why we selected these two peptides as high confidence starting points for the development of RBD – ACE2 PPI decoy peptides (Lines 275-290). Selection is initially derived from data that exists within the public domain – i.e. existing solved co-crystal structures and related publications that have already carried out extensive analysis. I have referenced these in text appropriately.

As analysis regarding contact residues between SARS-CoV-2 RBM and ACE2 has already been extensively investigated and published (references already included in manuscript), I have not included data in Figure 1 that details my own analysis as it would be directly replicating data that is already available and therefore add no novel value to the figure/study. As such, Figure 1 remains the same – highlighting the known contact residues of SARS-CoV-2 RBM that directly interact with ACE2. I have also highlighted these residues (in red) in a published (solved) co-crystal structure of the SARS-CoV-2 RBD – ACE2 protein-protein interaction for qualitative purposes (visualisation). This structure can be found on PDB and the protein structure I have used in the figure was generated using PyMol software. I hope this suffices.

c. Regarding comments RE: showing luminescence experimental set up is sound…

Having discussed this with Dr Brian Hudson and Dr Mario Rossie (the two researchers responsible for developing and carrying out the luciferase assay), they have forwarded the following response to your comment(s): 

‘We appreciate the reviewers’ comment, however we would also like to highlight that these internalization approaches based on the pH dependence of the NanoLuc activity have already been tested extensively by the scientific community. These tools have in fact been considered well established, valuable approaches for quantification of cell membrane protein internalization, especially in HEK293 cell lines. In the resubmitted version of the paper, you will find two new reference articles showing nano luciferase as a sensitive bioluminescent reporter to measure quantitatively the internalization of cell membrane receptors.

“Quantitative measurement of cell membrane receptor internalization by the nanoluciferase reporter: Using the G protein-coupled receptor RXFP3 as model. Y. Liu et al. / Biochimica et Biophysica Acta 1848 (2015) 688–694”, “A luminescent assay for real-time measurements of receptor endocytosis in living cells. M.B. Robers et al. / Analytical Biochemistry 489 (2015) 1e82“’

I hope this amendment/response will suffice.

(8) Response to Reviewer 2:

a. Sequences added to abstract as requested.

b. Although we did not generate any data regarding point mutations within our peptides that would confer higher ACE2 binding affinity, I have included a small section in the updated tracked changed document (Lines 406-410) briefly discussing the possibility of selecting for already known RBD point mutations that could lead to higher ACE2 binding affinity of our selected RBM peptides. Due to grant funding/duration coming to an end, we are unable to carry out further experiments relating to mutations. I hope this statement + relevant addition of references will suffice.

“Furthermore, analysis of global genome sequencing datasets detailing identified mutations within the receptor binding domain have identified variants that possess higher ACE2 binding affinity (e.g. Y453F and N501Y) compared with the original Wuhan-1 strain 33,34. Resultingly, these mutations could be leveraged in future efforts aimed at selecting peptide sequences that possess higher ACE2 binding affinity.”

---

## [Editor Report · Decision Letter 2]

8 Nov 2021

Peptides derived from the SARS-CoV-2 Receptor Binding Motif  bind to ACE2 but do not block ACE2-mediated host cell entry or pro-inflammatory cytokine induction.

PONE-D-21-28336R2

Dear Dr. Blair,

We’re pleased to inform you that your manuscript has been judged scientifically suitable for publication and will be formally accepted for publication once it meets all outstanding technical requirements.

Kind regards,

Tianwen Wang, Ph.D.

Academic Editor

PLOS ONE

Additional Editor Comments (optional):

The authors have responded to all the questions raised by reviewers sufficiently. No other revision is needed.
---

## [Editor Report · Acceptance letter]

10 Nov 2021

PONE-D-21-28336R2 

Peptides derived from the SARS-CoV-2 Receptor Binding Motif bind to ACE2 but do not block ACE2-mediated host cell entry or pro-inflammatory cytokine induction. 

Dear Dr. Blair:

I'm pleased to inform you that your manuscript has been deemed suitable for publication in PLOS ONE. Congratulations! Your manuscript is now with our production department. 

Kind regards, 

on behalf of

Dr. Tianwen Wang 

Academic Editor

PLOS ONE